# miRNA Expression May Have Implications for Immunotherapy in PDGFRA Mutant GISTs

**DOI:** 10.3390/ijms232012248

**Published:** 2022-10-14

**Authors:** Gloria Ravegnini, Margherita Nannini, Valentina Indio, Cesar Serrano, Francesca Gorini, Annalisa Astolfi, Aldo Di Vito, Fabiana Morroni, Maria Abbondanza Pantaleo, Patrizia Hrelia, Sabrina Angelini

**Affiliations:** 1Department of Pharmacy and Biotechnology, University of Bologna, 40126 Bologna, Italy; 2Department of Specialized, Experimental and Diagnostic Medicine, Sant’Orsola-Malpighi Hospital, University of Bologna, Via Massarenti 9, 40138 Bologna, Italy; 3Division of Oncology, IRCCS Azienda Ospedaliero Universitaria di Bologna, 40138 Bologna, Italy; 4Department of Veterinary Medical Sciences, University of Bologna, 40164 Ozzano, Italy; 5Sarcoma Translational Research Laboratory, Vall d’Hebron Institute of Oncology (VHIO), Vall d’Hebron Hospital Campus, C/ Natzaret 115-117, 08035 Barcelona, Spain; 6Department of Medical Oncology, Vall d’Hebron University Hospital, P/Vall d’Hebron 119, 08035 Barcelona, Spain; 7Inter-Departmental Center for Health Sciences & Technologies, CIRI-SDV, Alma Mater Studiorum-University of Bologna, 40126 Bologna, Italy

**Keywords:** gastrointestinal stromal tumors, GISTs, PDGFRA, D842V, miRNAs, microRNAs

## Abstract

Gastrointestinal stromal tumors (GISTs) harboring mutations in the *PDGFRA* gene occur in only about 5–7% of patients. The most common *PDGFRA* mutation is exon 18 D842V, which is correlated with specific clinico-pathological features compared to the other *PDGFRA* mutated GISTs. Herein, we present a miRNA expression profile comparison of *PDGFRA* D842V mutant GISTs and *PDGFRA* with mutations other than D842V (non-D842V). miRNA expression profiling was carried out on 10 patients using a TLDA miRNA array. Then, miRNA expression was followed by bioinformatic analysis aimed at evaluating differential expression, pathway enrichment, and miRNA-mRNA networks. We highlighted 24 differentially expressed miRNAs between D842V and non-D842V GIST patients. Pathway enrichment analysis showed that deregulated miRNAs targeted genes that are mainly involved in the immune response pathways. The miRNA-mRNA networks highlighted a signature of miRNAs/mRNA that could explain the indolent behavior of the D842V mutated GIST. The results highlighted a different miRNA fingerprint in *PDGFRA* D842V GISTs compared to non-D842Vmutated patients, which could explain the different biological behavior of this GIST subset.

## 1. Introduction

Gain-of-function mutations *KIT* and *PDGFRA* tyrosine kinases (TK) are responsible for between 85 and 90% of gastrointestinal stromal tumors (GIST), with the latter occurring only in approximately 5–10% of cases [1]. *PDGFRA* mutations commonly arise in exon 18 (~5%), with the Asp842Val (D842V) amino acid change accounting for more than 50% of *PDGFRA*-mutated GIST. More rarely, *PDGFRA* mutations emerge in the juxtamembrane domain encoded by exon 12 (~1%) or the ATP-binding domain coded by exon 14 (<1%) [1,2]. Irrespective of the exon, *KIT* or *PDGFRA* mutations lead to constitutive activation of these receptors and downstream signals, including PI3K/AKT/mTOR and RAS/RAF/MAPK, both essential for cell proliferation and survival [2,3,4,5]. Despite sharing downstream signaling, the variety of *KIT/PDGFRA* mutants across their exonic sequences results in different prognoses and responses to standard therapies with TK inhibitors (TKIs). Specifically, *PDGFRA* D842V mutant GISTs (referred to as D842V GISTs) represented, until recently, one of the major unmet needs in GIST clinical management. Indeed, this mutation confers primary resistance to first-line imatinib, and no proven efficacy is reported for the other approved TKIs [6,7,8,9,10,11,12]. In fact, D842V GISTs were considered orphan drugs until January 2021, when avapritinib demonstrated unprecedented clinical activity in this subgroup of patients, thereby achieving FDA regulatory approval [13,14,15]. Despite the recent achievement in the treatment of *PDGFRA* D842V GISTs, the therapeutic armamentarium for this rare subgroup remains limited, as there are no other effective treatments beyond avapritinib progression. In particular, the failure of identifying actionable events other than the *PDGFRA* D842V mutation prevents the identification of novel targets and, therefore, new drug development [16]. In addition, genetic alterations, epigenetic mechanisms play a key role in disease biology by controlling gene expression at the post-transcriptional level [3,17]. In this context, research interest in miRNAs has quickly increased and several studies have investigated the role of miRNAs in GIST development, classification, and prognosis [18,19,20]. However, most of the studies focused on *KIT* mutants or *KIT/PDGFRA* wild-type (WT) GISTs, leaving the molecular biology of *PDGFRA*-mutant GISTs largely unexplored. In view of these considerations, we performed a miRNA expression analysis in *PDGFRA* D842V versus *PDGFRA* non-D842V GISTs. Additionally, we integrated the results with the gene expression profile (GEP) [21] of the same sample set, previously published, in order to construct an original miRNA-mRNA regulatory network that may provide a significant contribution to future investigations, aiming at the identification of novel targets.

## 2. Results

### 2.1. Differential miRNA Expression between PDGFRA D842V Mutant versus PDGFRA Non-D842V Mutant GIST

The principal component analysis (PCA) did not show a clear difference between the two subgroups (Appendix A). However, the array highlighted 24 deregulated miRNAs out of the 768 analyzed. In particular, 10 miRNAs were upregulated and 14 were downregulated in D842V GISTs compared to non-D842V GISTs. All the differentially expressed miRNAs are reported in Table 1.

Hierarchical clustering of all samples separated *PDGFRA* D842V GISTs and non-D842V GISTs into two clusters (Figure 1).

Pathway enrichment analysis of the 24 deregulated miRNAs was performed using the miRNet tool; functional enrichment analysis is based on gene ontology (GO) terms or pathways according to Reactome databases [22]. Results are reported in the Appendix A; interestingly, we observed that deregulated miRNAs targeted genes mainly involved in the immune response pathways (Table 2 and Figure 2).

### 2.2. miRNA and mRNA Arrays Network

Among the 24 deregulated miRNAs, six had experimentally verified associations with their targets. Amongst the target genes, we highlighted the existence of three potential regulatory networks: miR-9-5p → BCL6 and SIRT1, miR-133B → SP1, and miR-210-3p → NPTX1. Networks are depicted in Figure 3.

## 3. Discussion

*KIT* and *PDGFRA* genotypes in GISTs are associated with clinicopathological prognostic or predictive features, including response to TKIs, thus underscoring intrinsic biological differences between the diversity of driver mutations [23]. However, it has yet to be understood the underlying biological mechanisms behind specific clinicopathological features and the *KIT/PDGFRA* genotype. This sea of uncertainty precludes the opportunity to develop novel drugs, which is of particular importance for D842V GISTs. Indeed, even if TKIs have revolutionized the treatment of advanced metastatic or unresectable GISTs [17], D842V GISTs are primarily resistant to imatinib, and based on targeted kinase inhibition. Currently, the only available pharmacological option for this subset of GISTs is the recently approved avapritinib. However, after the onset of resistance, there are no treatment alternatives. In this scenario, investigating the epigenetic landscape may provide novel insights into the biology of this specific GIST molecular subgroups with potential implications for GIST therapeutics. Based on this, comparison of miRNA expression profiles between *PDGFRA* D842V and non-D842V mutant-GISTs was undertaken for the first time. To further deepen into the biology of PDGFRA-mutant GISTs, we applied bioinformatics tools to generate miRNA-mRNA networks, using GEP data from the same sample set publicly available [21]. The generation of miRNA-mRNA networks may represent a promising approach to develop specific interventions and the selection of potential pharmacological targets in *PDGFRA*-mutant GISTs.

Notably, *PDGFRA*-mutant GISTs are characterized by the strongest immune-signature and immune-pathway enrichment compared to other GIST molecular subtypes, including *KIT* mutant and SDH-deficient GISTs [23,24,25]. Conversely, our study supports that D842V GISTs represent a tumor subset with distinctive biological behavior compared to other GISTs with non-D842V *PDGFRA* mutations. Indeed, *PDGFRA* D842V GISTs display a different miRNA signature compared to non-D842V GISTs. These results, in turn, agree with our prior GEP analysis [21]. Specifically, both GEP and miRNA analyses highlighted that, within the *PDGFRA* mutant GISTs, the D842V subgroup shows a distinctive immunological fingerprint. This could explain why, to date, despite the involvement of the immune system in controlling the disease, the results of immunotherapy are disappointing [26]. In view of these differences, it is likely that some GISTs harboring unique molecular backgrounds may benefit from specific immunotherapeutic approaches, and the identification of new molecular targets can have translational implications. The results of our miRNA profiling highlighted several miRNAs already reported in the literature as involved in immunity, including miR-149-3p, miR-33a-5p, and miR-15a. Among the most significantly deregulated miRNAs, two were downregulated—miR-708-5p and miR-1225p—, and one—miR-431-3p—was upregulated in D842V GISTs compared to the other *PDGFRA* mutations. None of these miRNAs has been previously reported as deregulated in GISTs. This could be due to the lack of patient stratification based on molecular subgroups, which was the primary scope of this study.

miR-708-5p and miR-122-5p function as tumor suppressors in multiple cancer types, including colorectal and gastric cancers [27,28,29]. Their downregulation appears to be associated with a lncRNA/circRNA-mediated mechanism of miRNA sponge. In this context, lncRNAs and circRNAs analysis can be an emerging area of research in GISTs. Further investigations are needed to deepen the biological role and therapeutic potential of these two miRNAs, in particular in D842V GIST patients. Intriguingly, miR-431-5p is significantly upregulated in D842V GISTs, while available data ascribes a tumor-suppressor role to this miRNA. Considering the clinico-pathological peculiarities of the D842V group compared to other *PDGFRA*-mutant GISTs, together with its uniqueness, this finding deserves further investigation.

With regard to the miRNA-mRNA networks, we highlighted the involvement of three different miRNAs—miR-9-5p, miR-133b, and miR-210-5p—targeting four different genes: BCL6, SIRT1, SP1, and NPTX1. Validated targets of miR-9-5p are BCL6, a nuclear protein with transcriptional repressor activity [30], and SIRT1, a class III histone deacetylase having multiple functions in cancer progression [26]. Specifically, BCL6, was previously found highly expressed in mesenchymal tumors, including GISTs, with a mechanism unrelated to rearrangements in the *BCL6* gene [31]. Our study suggests that BCL6 overexpression could depend on an epigenetic mechanism [21]. With regard to SIRT1, its role in cancer remains controversial, although its overexpression has been associated with poor prognosis in several cancer types [32,33,34]. Therefore, its downregulation in D842V GISTs could explain, at least in part, the indolent nature of these tumors in a localized setting, compared to other GIST molecular subgroups. Notably, both miR-9-5p targets have a role in immunity. In particular, BCL6 is induced by IL4 and IL21, a potent B-cell growth and differentiation factor, able to induce Ig proliferation and class switching, as well as the production of large quantities of secreted Ig [35,36]. On the other hand, SIRT1 has recently emerged as a regulator of both innate and adaptive immune responses through different mechanisms [28]. Therefore, miR-9-5p, together with its targets, may influence the tumor micro-environment composition.

Regarding miR-133b, three different studies in GISTs showed its downregulation [37,38,39]; by contrast, our analysis revealed that miR-133b is upregulated in D842V GISTs. This could be due to the limited sample size, which is a limitation. However, that also affects the other reports, with the number of patients spanning from 8 to 53, mostly due to the rarity of GIST. Nevertheless, none of the previous studies focused on *PDGFRA* mutant GISTs, with most of the cases harboring KIT primary mutations. Therefore, it is possible that miR-133b upregulation could represent a peculiar trait of D842V GISTs, an aspect which is also supported by GEP results. Indeed, the downregulation of the validated target, SP1, together with the predicted PPP2R2D, ZHX3, and CREB5, was identified. Remarkably, all these genes may participate in the transcription of many important regulatory genes correlated with cancer development and progression. Therefore, their downregulation in D842V GISTs may also explain the indolent behavior of this subgroup of GISTs [33,34,35,36].

With regard to miR-210-3p, it is known to be involved in several human cancers, even with some controversial results [40,41]. This miRNA has never been reported in GISTs, therefore, even if we cannot exclude a chance-finding, we hypothesized that it is a peculiar trait of D842V GISTs. However, the observed miR-210-3p downregulation was supported by the upregulation of its target, NPTX1, identified in our previous GEP analysis. A recent study ascribes a metastasis-related significance to this miRNA [40], also known as hypoxamiR, due to its role in cellular response to hypoxia, which includes angiogenesis, cell proliferation, and differentiation [42].

## 4. Materials and Methods

### 4.1. Patients

Clinical and pathological characteristics of the 10 patients included in this study have been extensively described elsewhere [21]. In brief, specimens were freshly collected from untreated GIST localized in the stomach; all patients had a localized disease status. Table 3 summarizes the main patients’ characteristics.

This study was approved by the local Institutional Ethical Committee of Azienda Ospedaliero-Universitaria Policlinico S.Orsola-Malpighi (number 113/2008/U/Tess). The GIST diagnosis was confirmed by expert pathologists through histological re-evaluation and immunohistochemistry for CD117 and DOG1. All patients harbored gain-of-function mutations in the *PDGFRA* gene. Specifically, 5 patients had PDGFRA exon 18 D842V mutation and 5 had *PDGFRA* non-D842V mutations (*n* = 3 in exon 12, *n* = 1 in exon 14, and *n* = 1 in exon 18). Mutational analysis of *KIT* (exons 8, 9, 11, 13, and 17) and *PDGFRA* (exons 12, 14, and 18) genes was performed on genomic DNA extracted from paraffin-embedded (FFPE) tumor tissue [21]. Total RNA was extracted from fresh frozen tumor specimens using RNeasy Mini Kit (Qiagen, Milan, Italy).

### 4.2. miRNA Expression Profiling

miRNA profiling was performed using TaqMan Low Density Arrays (TLDA) and pools A and B (Applied Biosystems, Waltham, MA, USA), which allow to analyze 768 miRNAs. Total RNA was retrotranscribed through the TaqMan MicroRNA Reverse Transcription Kit (Applied Biosystems) and MegaPlex RT primers (Applied Biosystems) pools A and B. cDNAs were preamplified using TaqMan PreAmp Master Mix and PreAmp primers (pools A and B; Applied Biosystems). The miRNA arrays were loaded with the preamplified samples and run on a 7900HT Fast Real-Time PCR System (Applied Biosystems).

### 4.3. Bioinformatic Analysis

miRNA data were analyzed with SDS Relative Quantification Software version 2.4. (Applied Biosystems); miRNAs with Ct values ≥ 35 were considered as not expressed and were excluded from further analysis. Normalization was carried out by subtracting the U6 mean Ct from individual Ct values. The R-Bioconductor package Limma was adopted to evaluate the differential expression profile between the D842V and non-D842V GISTs. Heatmaps were generated using the Multiple Experiment Viewer (MEV) tool. Analysis of the principal component (PCA) was developed by the R CRAN package rgl and visualized through the Cubemaker online tool. Deregulated miRNAs were analyzed using the miRNet tool (https://www.mirnet.ca/miRNet/home.xhtml (accessed on 1 September 2022); pathway enrichment analysis was performed with the Function Explorer module (Database Reactome) and gene ontology was explored with the same module; the software uses standard enrichment analysis based on the hypergeometric tests after adjustment for false discovery rate. miRNet integrates data from publicly available different miRNA databases such as TarBase, miRTarBase, and miRecords and allows users to construct miRNA-target interaction networks at different confidence levels [43]. Subsequently, to build potential miRNA-mRNA networks, we used GEP data previously generated from the same set of patients [21].

## 5. Conclusions

To the best of our knowledge, this is the first work to analyze miRNA profiles in two subgroups of GIST patients with different mutations in *PDGFRA* genes. The present study highlights a different miRNA fingerprint in *PDGFRA* D842V GISTs compared to the other *PDGFRA* mutated patients, which could explain the indolent behavior of this GIST subset. Interestingly, our finding is consistent with previous work on the same sample set, highlighting a prominent immunological signature and a lack of an oncogenic signature in the D842V group.

Aware of the limitations (i.e., the small sample size), this study underlines the importance of patient stratification based not only on tumor genetics but also on epigenetic fingerprints. This multifaced approach (i.e., stratification according to all the features potentially contributing to tumorigenesis) could be of help in identifying further modalities for targeting the disease.

## Figures and Tables

**Figure 1 ijms-23-12248-f001:**
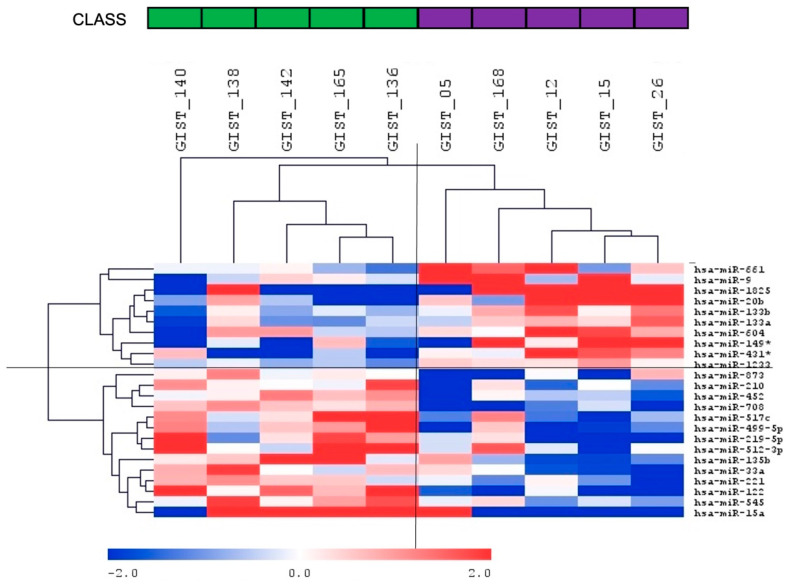
miRNA hierarchical clustering. The heatmap shows miRNAs identified as differentially expressed between *PDGFRA* D842V mutant GISTs compared to non-D842V mutant GISTs. * indicates 3p (i.e., miR-149-3p; miR-431-3p).

**Figure 2 ijms-23-12248-f002:**
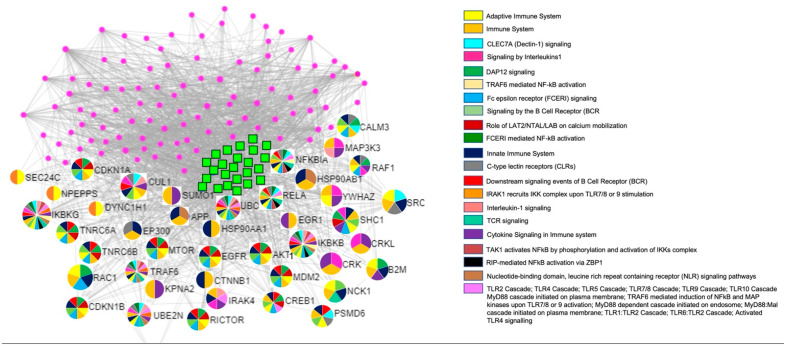
Main genes involved in immune pathways. The green squares represent the 24 deregulated miRNAs; each miRNA target gene is represented with a multicolored circle, based on the specific immune pathways in which it is involved. Pink dots represent other target genes not involved in immune pathways.

**Figure 3 ijms-23-12248-f003:**
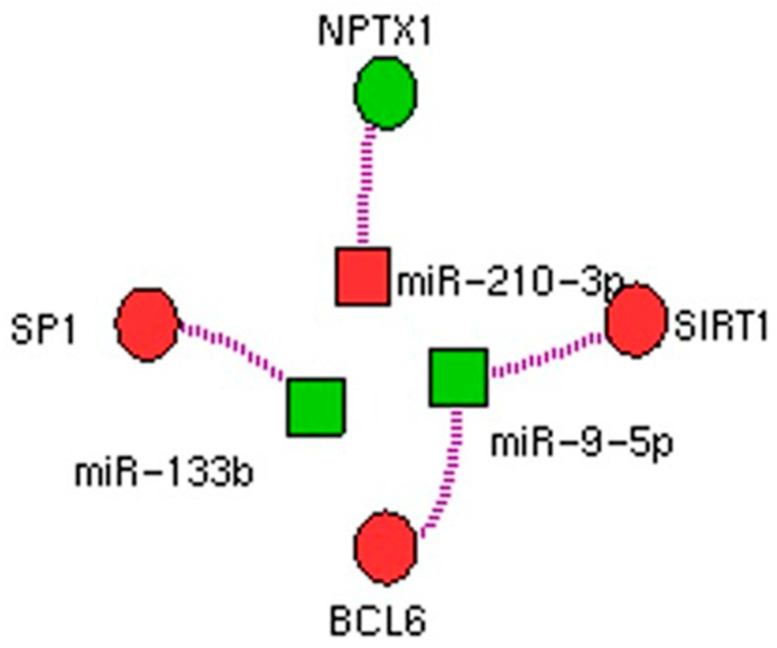
Networks between miRNAs and genes: deregulated miRNAs and deregulated genes in PDGFRA D842V GISTs vs. non-D842V GISTs. Green color represents up-regulation and red color represents down-regulation.

**Table 1 ijms-23-12248-t001:** Deregulated miRNAs identified comparing D842V vs. non-D842V GIST patients.

miRNA ID	Delta Ct	D842V vs. Non-D842V	*p*-Value	Predicted Targets among the Deregulated Gene in Our Cohort of Patients *
hsa-miR-1825	−5.31	↑	0.027	NLK(2)
hsa-miR-431-3p	−4.96	↑	0.009	
hsa-miR-20b-3p	−3.87	↑	0.015	
hsa-miR-149-3p	−3.40	↑	0.037	SPRY4(2)
hsa-miR-9-5p	−2.98	↑	0.038	GNPNAT1(2), SIRT1(3), CREB5(2), POU2F1(3), BCL6(3), PXDN(3), RNF169(2), FBN2(3), PTAR1(2), NIN(2)
hsa-miR-604	−2.82	↑	0.044	
hsa-miR-661	−1.64	↑	0.049	IL17RA(2)
hsa-miR-133b	−1.43	↑	0.032	PPP2R2D(2), SP1(2), ZHX3(2), CREB5(2),
hsa-miR-133a-3p	−1.38	↑	0.044	
hsa-miR-1233-3p	−1.07	↑	0.042	
hsa-miR-545-3p	1.32	↓	0.048	TSPAN2(2)
hsa-miR-210-3p	1.56	↓	0.046	NPTX1(1)
hsa-miR-221-3p	1.59	↓	0.039	
hsa-miR-135b-5p	2.10	↓	0.027	
hsa-miR-33a-5p	2.12	↓	0.043	
hsa-miR-452-5p	2.18	↓	0.044	
hsa-miR-219a-5p	2.43	↓	0.019	
hsa-miR-499a-5p	2.72	↓	0.017	
hsa-miR-517c-3p	2.79	↓	0.024	
hsa-miR-873-5p	3.07	↓	0.027	
hsa-miR-512-3p	3.60	↓	0.032	
hsa-miR-708-5p	3.92	↓	0.002	
hsa-miR-122-5p	4.45	↓	0.010	CD320(2)
hsa-miR-15a-5p	6.43	↓	0.025	RSPO3(2)

↑: up-regulation; ↓: down-regulation; * in brackets: number of in silico tools predicting the same target which was also deregulated in previous gene expression analysis see ref. [21].

**Table 2 ijms-23-12248-t002:** Top 25 pathways related with immune system.

Pathway	Adjusted *p* Value
Fc epsilon receptor (FCERI) signaling	1.88 × 10^−11^
Signaling by the B Cell Receptor (BCR)	4.35 × 10^−10^
Innate Immune System	6.25 × 10^−10^
Downstream signaling events of B Cell Receptor (BCR)	7.55 × 10^−9^
Adaptive Immune System	1.82 × 10^−7^
TAK1 activates NFkB by phosphorylation and activation of IKKs complex	2.35 × 10^−6^
Toll Like Receptor 10 (TLR10) Cascade	2.51 × 10^−6^
Toll Like Receptor 5 (TLR5) Cascade	2.51 × 10^−6^
MyD88 cascade initiated on plasma membrane	2.51 × 10^−6^
TRAF6 mediated induction of NFkB and MAP kinases upon TLR7/8 or 9 activation	2.74 × 10^−6^
Toll Like Receptor 7/8 (TLR7/8) Cascade	2.92 × 10^−6^
MyD88 dependent cascade initiated on endosome	2.92 × 10^−6^
MyD88:Mal cascade initiated on plasma membrane	4.24 × 10^−6^
Toll Like Receptor TLR1:TLR2 Cascade	4.24 × 10^−6^
Toll Like Receptor TLR6:TLR2 Cascade	4.24 × 10^−6^
Toll Like Receptor 2 (TLR2) Cascade	4.24 × 10^−6^
IRAK1 recruits IKK complex	4.26 × 10^−6^
IRAK1 recruits IKK complex upon TLR7/8 or 9 stimulation	4.26 × 10^−6^
Cytokine Signaling in Immune system	4.27 × 10^−6^
TRAF6 mediated NF-kB activation	5.53 × 10^−6^
Toll Like Receptor 9 (TLR9) Cascade	3.67 × 10^−6^
CLEC7A (Dectin-1) signaling	1.29 × 10^−6^
Interleukin-1 signaling	1.29 × 10^−6^
Role of LAT2/NTAL/LAB on calcium mobilization	1.81 × 10^−6^
DAP12 signaling	1.09 × 10^−4^

**Table 3 ijms-23-12248-t003:** Patients’ characteristics.

Patient ID	Size (cm)	Mitotic Index (HPF) *	Last Follow Up ^§^	PDGRA Molecular Analysis
GIST140	15	3/50	AWOD	Exon 18 D842V
GIST165	12	2/50	AWOD	Exon 18 D842V
GIST138	7	8/50	AWOD	Exon 18 D842V
GIST142	3	5/50	AWOD	Exon 18 D842V
GIST136	4.5	6/50	DNFD	Exon 18 D842V
GIST05	7	4/50	AWOD	Exon 12 del 16117-20 CCCG + ins 16124 TC + del 16124-30 GGACATG
GIST12	NA	NA	NA	Exon 14 K646E
GIST15	NA	NA	NA	Exon 18 DIMH842-845del
GIST26	NA	NA	NA	Exon 12 V561D
GIST168	5.5	4/50	AWOD	Exon 12 S566_E571 > R

***** 50× High Power Field. **^§^** AWOD, alive without disease; DNFD, dead not for disease. NA: not available information.

## Data Availability

The original contributions presented in the study are publicly available. These data can be found at https://www.ncbi.nlm.nih.gov/geo/.

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
