# Peer review of "miRNA Expression May Have Implications for Immunotherapy in PDGFRA Mutant GISTs"

_ijms, 2022, doi:10.3390/ijms232012248_

Round 1

Reviewer 1 Report

N/A

Author Response

we thank the reviewer

Reviewer 2 Report

This study is well written and data are presented clearly, although the small number of samples evaluated is a major weakness.

It would be interesting to see if the two group of tumors/patients had different clinical pathological features and disease course: authors should include more clinical and pathological data (tumor size, mitotic index, therapy, clinical evolution….).

Because PDGFRA-mutated GISTs are characterized by a strong immune signature enrichment, it would be interesting to try to consider if these differentially expressed microRNA may be expressed by immune cells within the tumor. Previously, authors analyzed tumor microenvironment composition in these same samples. Is there any correlation between immune infiltrate and microRNA expression in these samples?

Table 3: mutation K642E is in exon 14, (not exon 18).

Author Response

This study is well written and data are presented clearly, although the small number of samples evaluated is a major weakness.

RESPONSE. We thank the reviewer for positive overview. All the changes have been highlighted in red

It would be interesting to see if the two group of tumors/patients had different clinical pathological features and disease course: authors should include more clinical and pathological data (tumor size, mitotic index, therapy, clinical evolution….).

RESPONSE. We thank the reviewer for positive comments. Clinical information was added part in the text and part in the revised table 3.

Because PDGFRA-mutated GISTs are characterized by a strong immune signature enrichment, it would be interesting to try to consider if these differentially expressed microRNA may be expressed by immune cells within the tumor. Previously, authors analyzed tumor microenvironment composition in these same samples. Is there any correlation between immune infiltrate and microRNA expression in these samples?

RESPONSE. We thanks the reviewer for the interesting comment. We added an insight in the conclusion section.

Table 3: mutation K646E is in exon 14, (not exon 18).

RESPONSE. We Thank the reviewer, the mistake has been corrected

Reviewer 3 Report

I appreciate the authors for the efforts on contribution of article entitled "miRNA expression may have implications for immunotherapy in PDGFRA mutant GISTs"

I suggest the authors to look into the recent publications on similar area of research and correlation of the work presented and how the current work is novel and advantage over the previous work. 

Example: Integrated Antitumor Activities of Cellular Immunotherapy with CIK Lymphocytes and Interferons against KIT/PDGFRA Wild Type GIST : Int. J. Mol. Sci. 202223(18), 10368; https://doi.org/10.3390/ijms231810368

Abstract: Can be improved by adding the results and presenting in systematic way, Aim/scope of the current work, Methods with Results and Conclusion

Did the authors validate the Results from the software? How many times the experiment was performed to validate the data.

Figure 2. Main genes involved in immune pathways. Image quality can be improved

Did the authors check the regulation of MicroRNA-494 in this context? if not what is the reason for not considering this?

Author Response

I appreciate the authors for the efforts on contribution of article entitled "miRNA expression may have implications for immunotherapy in PDGFRA mutant GISTs"

RESPONSE: we thank the reviewer for the positive comment

I suggest the authors to look into the recent publications on similar area of research and correlation of the work presented and how the current work is novel and advantage over the previous work. -Example: Integrated Antitumor Activities of Cellular Immunotherapy with CIK Lymphocytes and Interferons against KIT/PDGFRA Wild Type GIST: Int. J. Mol.Sci. 2022, 23(18),10368;https://doi.org/10.3390/ijms231810368

RESPONSE. We are not aware of novel/recent literature focused on PDGFRA mutated GIST only. We would be happy to include the work suggested by the reviewer, however, we found difficult to include it in our work. The work suggested by the reviewer is on KIT/PDGFRA wild type GIST, while our work has been performed on PDGFRA mutant GIST. Therefore, to the best of our knowledge, the results we provided are totally novel, and add insight to a specific subgroup of GIST patients, that are the PDGFRA mutated one. We would appreciate if the reviewer could provide us with is key to the reading, so that we could include the suggested work.

Abstract: Can be improved by adding the results and presenting in systematic way, Aim/scope of the current work, Methods with Results and Conclusion

RESPONSE. The abstract has been modified according to the Author’s suggestion.

Did the authors validate the Results from the software? How many times the experiment was performed to validate the data.

RESPONSE. Unfortunately, only technical replicates could be performed. Indeed, given the small amount of tumor sample collected at the moment of tumour resection and considering the multiple analyses needed (i.e DNA, RNA isolation, diagnostic characterization, etc) we could not repeat the experiment multiple times. Therefore, it was not possible to have a real biological replicate. However, considering that the expression of target genes is in agreement with the expression of the miRNAs, we believe that the result of the profiling (that had technical replicates) is closed to reality

Figure 2. Main genes involved in immune pathways. Image quality can be improved

RESPONSE. The figure is already at the maximum of the pixels allowed by the program used to generate it. However, we tried to rotate the figure and this should improve its readability

Did the authors check the regulation of MicroRNA-494 in this context? if not what is the reason for not considering this?

RESPONSE. miR-494 was not deregulated in our sample set. This is not surprising considering that this miRNA has been mainly implicated in the regulation of KIT expression, whilst the samples we used were all PDGFRA mutated.